# Chassis-based fiber-coupled optical probe design for reproducible quantitative diffuse optical spectroscopy measurements

Giselle C. Matlis[1,2], Qihuang Zhang[3,4], Emilie J. Benson[1,5], M. Katie Weeks[6], Kristen Andersen[1], Jharna Jahnavi[1], Alec Lafontant[1], Jake Breimann[1], Thomas Hallowell[6], Yuxi Lin[6], Daniel J. Licht[1,7,8,9], Arjun G. Yodh[5], Todd J. Kilbaugh[6,7], Rodrigo M. Forti[1], Brian R. White[7,10], Wesley B. Baker[1,7], Rui Xiao[3,7], Tiffany S. Ko[6]*

1 Division of Neurology, Department of Pediatrics, Children's Hospital of Philadelphia, Philadelphia, PA, United States of America, 2 School of Biomedical Engineering, Science and Health Systems, Drexel University, Philadelphia, Pennsylvania, United States of America, 3 Department of Biostatistics and Epidemiology, University of Pennsylvania, Philadelphia, PA, United States of America, 4 Department of Epidemiology, Biostatistics and Occupational Health, McGill University, Montreal, QC, Canada, 5 Department of Physics and Astronomy, University of Pennsylvania, Philadelphia, PA, United States of America, 6 Department of Anesthesiology and Critical Care Medicine, Children's Hospital of Philadelphia, Philadelphia, PA, United States of America, 7 Perelman School of Medicine at the University of Pennsylvania, Philadelphia, PA, United States of America, 8 Division of Neurology, Department of Pediatrics, Children's National, Washington, District of Columbia, United States of America, 9 Division of Neurology, George Washington University, Washington, District of Columbia, United States of America, 10 Division of Pediatric Cardiology, Department of Pediatrics, The Children's Hospital of Philadelphia, Philadelphia, PA, United States of America

* kotiff@chop.edu

**Data Availability Statement:** All relevant data are within the manuscript and its Supporting Information files.

## Abstract

Advanced optical neuromonitoring of cerebral hemodynamics with hybrid diffuse optical spectroscopy (DOS) and diffuse correlation spectroscopy (DCS) methods holds promise for non-invasive characterization of brain health in critically ill patients. However, the methods' fiber-coupled patient interfaces (probes) are challenging to apply in emergent clinical scenarios that require rapid and reproducible attachment to the head. To address this challenge, we developed a novel chassis-based optical probe design for DOS/DCS measurements and validated its measurement accuracy and reproducibility against conventional, manually held measurements of cerebral hemodynamics in pediatric swine (n = 20). The chassis-based probe design comprises a detachable fiber housing which snaps into a 3D-printed, circumferential chassis piece that is secured to the skin. To validate its reproducibility, eight measurement repetitions of cerebral tissue blood flow index (BFI), oxygen saturation ($StO_2$), and oxy-, deoxy- and total hemoglobin concentration were acquired at the same demarcated measurement location for each pig. The probe was detached after each measurement. Of the eight measurements, four were acquired by placing the probe into a secured chassis, and four were visually aligned and manually held. We compared the absolute value and intra-subject coefficient of variation (CV) of chassis versus manual measurements. No significant differences were observed in either absolute value or CV between chassis and manual measurements (p > 0.05). However, the CV for BFI (mean ± SD: manual, 19.5% ± 9.6; chassis, 19.0% ± 10.8) was significantly higher than $StO_2$ (manual, 5.8% ±

**Funding:** This work was financially supported by the National Institutes of Health (NIH) grant numbers K08-NS117897 (BRW), R01-NS113945 (GCM, AL, WBB), R01-HL141386 (MKW, TH, YL, TJK), T32-HL007915 (TSK), and P41-EB015893 (AGY, WBB). This work was also financially supported by the Children's Hospital of Philadelphia Frontier Program (GCM, EJB, KA, JJ, JB, YL, DJL, TJK, RMF, WBB, RX, TSK). This work was also financially supported by the American Heart Association (AHA) grant number 24SCEFIA1260971 (TSK). The funders had no role in study design, data collection and analysis, decision to publish, or preparation of the manuscript.

**Competing interests:** The authors have declared that no competing interests exist.

6.7; chassis, 6.6% ± 7.1) regardless of measurement methodology (p<0.001). The chassis-based DOS/DCS probe design facilitated rapid probe attachment/re-attachment and demonstrated comparable accuracy and reproducibility to conventional, manual alignment. In the future, this design may be adapted for clinical applications to allow for non-invasive monitoring of cerebral health during pediatric critical care.

## Introduction

Application of advanced, non-invasive diffuse optical neuromonitoring of cerebral hemodynamics in critical care has shown promise as an indicator of neurological outcomes that can be used to guide patient management [1–10]. Critical care scenarios are emergent and require rapid application of monitoring devices. Commercial continuous-wave (CW) near-infrared spectroscopy (NIRS) cerebral oximeters have been widely adopted due to clinical interest in brain monitoring and ease of use [11–19]. These devices typically employ light-emitting diode sources and compact photodiode detectors that are packaged within a lightweight patient interface and are secured using skin-safe adhesives. However, there remains limited consensus or standardized guidance for data interpretation, in part due to concerns over reproducibility and accuracy [12, 20–22].

To address these limitations, advanced diffuse optical techniques have been applied and have demonstrated clinical utility [9]. These advanced diffuse optical measurement techniques include broadband (or hyperspectral, bDOS), frequency-domain (FD-DOS), and time-domain diffuse optical spectroscopy (TD-DOS) for quantification of cerebral oxygenation and blood volume, as well as diffuse correlation spectroscopy (DCS) for quantification of cerebral blood flow and intracranial pressure. Concurrent DOS/DCS measurements further enable characterization of cerebral oxygen metabolism [23].

A challenge for current hybrid DOS/DCS devices concerns optical fiber coupling of patient measurement interfaces (probes) to optical sources and detectors. The use of fibers facilitates the use of more complex DOS/DCS instrumentation that cannot be placed directly on the head, however, this results in probes which are heavier and have larger form factors than in NIRS devices due to their stiffness and weight. Attachment of DOS/DCS probes is especially challenging for emergent applications such as informing intra-arrest interventions for cardiopulmonary resuscitation [2, 3, 24, 25]. Poor attachment impacts measurement accuracy and reproducibility; these have been identified as key obstacles to the use of diffuse optical devices for clinical management [2, 3, 24, 25].

In this work, we address the challenge of improving the reproducibility and ease of probe attachment in the setting of neuromonitoring. Existing fiber optic probes for advanced DOS and/or DCS have largely relied on the use of tape, wraps, or glue for head attachment which cannot be easily reproduced between longitudinal monitoring sessions [26–34]. Reduced within subject reproducibility, in turn, impacts the specificity and sensitivity of physiologic measurements for prediction of clinical outcomes [35, 36]. Herein, we present the design and validation of a novel chassis-based fiber optic probe. The primary objective of the design was to permit rapid, reproducible, and secure attachment and detachment of the optical probe from the same measurement location. A secondary objective was to validate that the novel probe design provided comparable accuracy and measurement reproducibility to conventional (manual) probe placement by visual alignment.

## Methods

### Design objectives and requirements

We designed and optimized a fiber optic probe design for non-invasive diffuse optical measurements of cerebral hemodynamics in pre-clinical piglet models of pediatric critical care [8, 23]. The final design concept consists of a detachable FD-DOS/DCS fiber housing probe which connects via magnets, screws, and divots to a circumferential chassis piece that is secured to the head. Herein, we describe the diffuse optical measurement techniques, associated design considerations and objectives, and finally, design considerations based on challenges identified from existing probes.

**Diffuse optical measurement techniques.** Assessment of the optical absorption and scattering properties of biological tissue permit non-invasive quantification of cerebral physiology by diffuse optical spectroscopies [9, 23]. In the present work, the optical probe was designed for compatibility with an existing clinical research instrument combining frequency-domain diffuse optical spectroscopy (FD-DOS) and diffuse correlation spectroscopy (DCS). The instrument used for this study was previously described [32].

The FD-DOS module advances upon commercially available continuous-wave (CW) near-infrared spectroscopy (NIRS) by using radio frequency-modulated sources (110 MHz for our device) to induce diffuse photon density waves in tissue; the detectors measure the light intensity amplitude and phase of these waves emerging from tissue at multiple source-detector distances. The amplitude and phase measurements versus source-detector distance are then fit to the homogeneous semi-infinite solution of the photon diffusion equation to extract both the optical tissue absorption and reduced scattering coefficients (note that in CW NIRS where there is no phase information, tissue reduced scattering is assumed) [37]. Tissue oxy- and deoxy-hemoglobin concentrations are derived using multispectral measurement of tissue absorption (i.e., at 690, 730, 786, and 830 nm wavelengths for our instrument) [38].

The DCS module leverages the time-dependent fluctuations in light intensity caused by the scattering of light off of moving red blood cells to non-invasively quantify tissue blood flow [8, 23]. Single-mode fibers coupled to single-photon counting avalanche photodiodes permit calculation of the intensity temporal autocorrelation function of a single speckle on the surface of the skin. The decay of this intensity autocorrelation function is proportional to the diffusion rate of moving molecules which scatter light (predominantly, red blood cells), and may be estimated using the semi-infinite correlation diffusion equation solution to obtain a blood flow index (BFI). BFI has been empirically shown to linearly correlate with cerebral blood flow measurements in children based on MRI and transcranial Doppler [8, 23, 39, 40].

### Design objectives and requirements

**Diffuse optical measurement design constraints and objectives.** Several constraints and objectives were applied to the design to allow for diffuse optical measurements of the brain. The choice of source-detector distances (SDS) is an important consideration for ensuring that the FD-DOS and DCS measurements have adequate depth-sensitivity to assess the brain. With typical tissue optical properties, the mean light penetration depth of the FD-DOS/DCS measurements is of order one-third to one-half of the SDS [23]. To access cerebral tissue in young children and in our pediatric swine model, where the distance from the skin's surface to the brain ranges from 0.5–1 cm, the typical SDS ranges from 1.5 to 3 cm [23, 41].

Our design also sought to minimize measurement error due to loss of skin contact, ambient light contamination, and tissue and probe motion. Loss of contact between the probe's surface and skin is commonly caused by subject movement. When it occurs, resultant contamination

in FD-DOS/DCS measurements from ambient light and source light that has not traveled through tissue can lead to noisy data, discontinuous data shifts, and detector saturation [26, 42]. Inconsistent contact also worsens motion artifacts in the data and hinders measurement reproducibility [26, 43, 44]. Thus, design features were implemented to prevent probe body motion within the chassis and to ensure contact of the probe body with the skin. Additional solutions were explored to prevent light leakage through the probe body, including light-blocking fiber housings inside the probe and light-absorbing probe materials.

Finally, to accommodate comparability of optical measurements acquired at various experimental timepoints where intermittent probe removal was necessary, a primary objective of our design was to facilitate rapid and reproducible attachment and reattachment. Similar to the impact of the loss of probe contact or probe motion, reattachment of the probe to a different location than prior measurements reduces measurement reproducibility and results in discontinuous data shifts in time which are unrelated to measured physiology. The first step to achieving reproducible attachment was the design of a circumferential, detachable chassis which would maintain probe position without impeding subject mobility. The second step was adjusting the historical design of the probe body to fit within the chassis while maintaining historical design elements for enclosure and stability of optical fibers. Of particular importance were design elements to ensure reproducible alignment of the probe body within the chassis and to prevent probe body motion within the chassis, as mentioned above.

**Design constraints and objectives based on prior probes.** Historical probe use in our lab demonstrated the need for a small overall probe footprint, durability, and cost-effective manufacturing and assembly.

Constraints on the size of the probe and chassis assembly are imposed on the lower end by the required SDS and on the upper end by coincident experimental procedures that also occupied space on the head. Fig 1 illustrates the limited measurable area on the forehead of pediatric swine subjects (~ 4 weeks old) that remains sensitive to the brain. The sagittal sinus running along the midline is avoided due to tissue inhomogeneity which impacts the validity of the semi-infinite solution of the photon diffusion equation [41].

As a result of these limitations, the total footprint of the probe, including the chassis, was restricted to 44 mm x 23 mm (the size of our existing probe). The probe height also needed to be minimal in thickness to reduce material weight, which induces strain on probe attachment and reduces tolerance to motion. To this end, in our design and in historical probe designs,

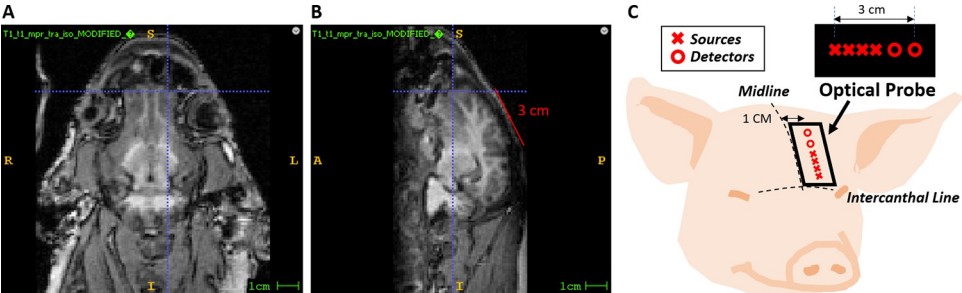

**Fig 1. Measurement area.** (a, b) A measurable area of a 4-week-old piglet brain from a superior and sagittal T1-weighted MRI, respectively, is shown. The horizontal blue line in (a) and (b) reflects the intercanthal line. The vertical line in (a) represents the approximate lateral position of the optical probe away from the midline, and the sagittal plane depicted in (b). Similarly, the vertical line in (b) corresponds to the superior plane depicted in (a). The red line in (b) reflects a 3 cm distance on the skin surface which corresponds to the longest source-detector separation. (c) Illustration of probe body positioning on the subject to ensure sources and detectors are overlying the brain for data collection.

90-degree bend fiber bundles were used at each source position and right-angle glass prisms were coupled to detector optical fibers bundles to permit the fiber bundles to lay parallel to the surface of the skin. Reduced thickness also allows for a wider range of subject positions and prevents procedure interference. Here, we imposed a maximal height of 15 mm.

Historical probe use in our lab demonstrated the need for (1) durability, specifically the need for strain-resistant suture points for subject attachment in the laboratory environment, and (2) ease of attachment, specifically the time and effort required to securely suture the probe. *Durability*: The existing probe design featured through-holes in the body of the optical probe to permit suture-reinforced attachment of the probe to the skin in preclinical subjects. Suture points on the probe were placed under constant mechanical strain during probe attachment. The suture points were the thinnest material regions yet had the most force applied; this led to breakage after repeated use. This breakage impeded subsequent efforts to secure the probe, which led to incomplete contact with the skin and, thus, ambient light contamination. In addition, this also often led to labor-intensive probe disassembly for repair (which often damaged the fiber optics) and/or costly probe replacement. Thus, strain and impact tolerance were extremely important factors in the physical geometry and material choice. This motivated the overall design goal to relocate the suture points from the probe body to a detachable chassis, thereby decoupling the integrity of the probe from that of the suture points. *Ease of Attachment*: The process of suturing the historical probe where attachment points were embedded within the body was labor intensive and tedious as it was difficult to visualize and access the end of the suture when threading through the probe body. Relocation of suture points to a detachable chassis significantly alleviates this issue. Additional considerations for suture point shape to facilitate suture alignment were explored by employing various geometries, including rods, squares, and triangular cutouts.

Finally, the design needed to be cost-effective in all aspects, from fabrication to ease of re-optimization. Material classes considered included metal, plastic, and rubber. Fabrication processes included CNC metal cutting, 3D printing, and injection molding. Metal cutting and injection molding did not allow for rapid revisions during prototyping. Our decision to use 3D printing allowed for rapid prototyping based on testing results and user input and was the most cost-effective. Once the probe and chassis designs were completed, optimal 3D printing materials were tested and chosen.

## Design, assembly, and testing

Our design process is illustrated in Fig 2. The final step assessed whether the chassis-probe design maintained or exceeded the accuracy and reproducibility of a hand-held optical probe.

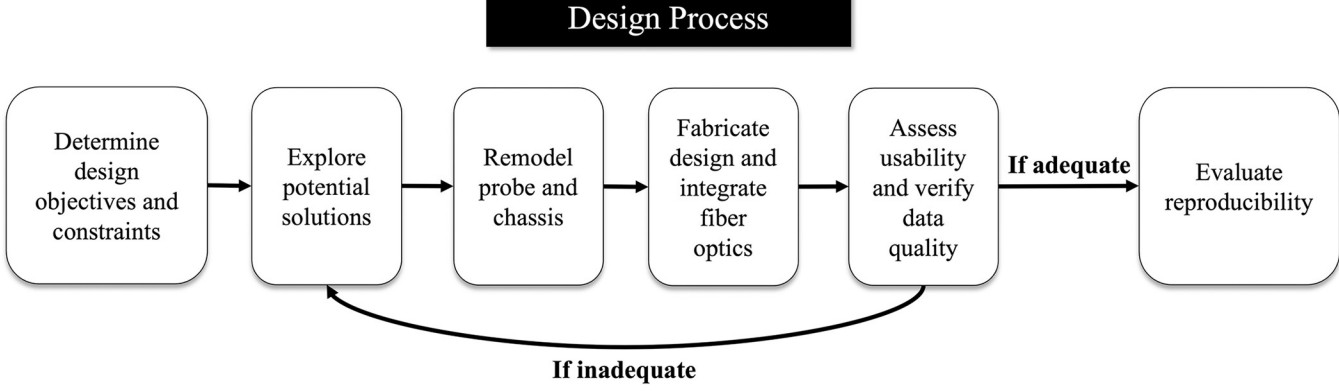

**Fig 2. Design process.** Flow chart demonstrating the design process used when creating the chassis-based probe design.

The optical probe and accompanying chassis were modeled in Google Sketchup 3D modeling software versions 2017 and 2019. The historical probe was designed as two pieces, a top and bottom piece, that enclosed and aligned the fibers; the bottom portion also included through-hole suture points. First, the historical probe was remodeled by removing excess material and suture points. Next, the chassis piece was designed around the base of the probe, and suture points were added. Based on the described design requirements, the probe and chassis design were optimized based on effectiveness and personnel feedback. After deciding on the use of 3D printing for probe manufacturing, prototype designs were printed using a Formlabs 2 printer (Formlabs, Somerville, Massachusetts) on a 10˚ angle, washed in 90% iso-propyl alcohol for 10–12 minutes to remove uncured resin, cleaned, and dried, and placed in a curing box for 10 minutes at 60˚ C. Once the individual pieces were cured and cleaned, they were assembled with the optical fibers and secured together with black liquid electrical tape (07315001126 LTB-400, Gardner Bender, New Berlin, Wisconsin) to complete probe body assembly.

## Design validation and assessment of measurement reproducibility

Once the chassis-based probe design was assembled, we assessed measurement reproducibility in pediatric swine. All procedures were approved by the CHOP Institutional Animal Care and Use Committee (IAC 21–001373, IAC 18–001318) and performed in strict accordance with the NIH Guide for the Care and Use of Laboratory Animals. General anesthesia was induced and maintained by 1–2% inhaled isoflurane, and all efforts were made to minimize pain and suffering. This protocol was conducted by three different individuals who were trained in optical measurement acquisition. The protocol consisted of eight distinct measurements on the left side of the subject's head—four "manual" measurements and four "chassis" measurements (Fig 3). Once a probe position with good data quality was identified, the location was outlined.

After the location was marked, "manual" measurements were acquired. The outlined location allowed for precise visual alignment of the probe and thus represents quantification of reproducibility under optimal conditions. Once the measurement was taken, the probe was then repositioned by being lifted from the head and placed back into the marked location. This method was repeated four times to complete "manual" collection. After the manual measurements, the detached chassis was sutured to the marked location on the subject's head. Once the chassis was sutured, the probe was placed into it for each "chassis" measurement. After each measurement was taken, the probe was repositioned by lifting it out of the chassis and placing it back. This method was repeated four times.

## Data and statistical analysis

MATLAB 2019b (Mathworks, Natick, Massachusetts) was used to process both the DCS and FD-DOS data to derive the Blood Flow Index (BFI), Tissue Oxygen Saturation (StO$_2$), Total Hemoglobin Concentration (THC), Oxygenated Hemoglobin ([HbO$_2$]) and Deoxygenated Hemoglobin ([Hb]) as previously described [32]. These data were compared between methods.

Three analyses were undertaken to (1) evaluate the accuracy of the chassis method as compared to the manual method, (2) evaluate the reproducibility of the chassis method as compared to the manual method, and (3) evaluate the reproducibility of FD-DOS StO$_2$ versus DCS BFI.

To evaluate if there was a significant difference between manual and chassis measurements, a two-sided paired $t$-test was used to compare the mean manual versus the mean chassis

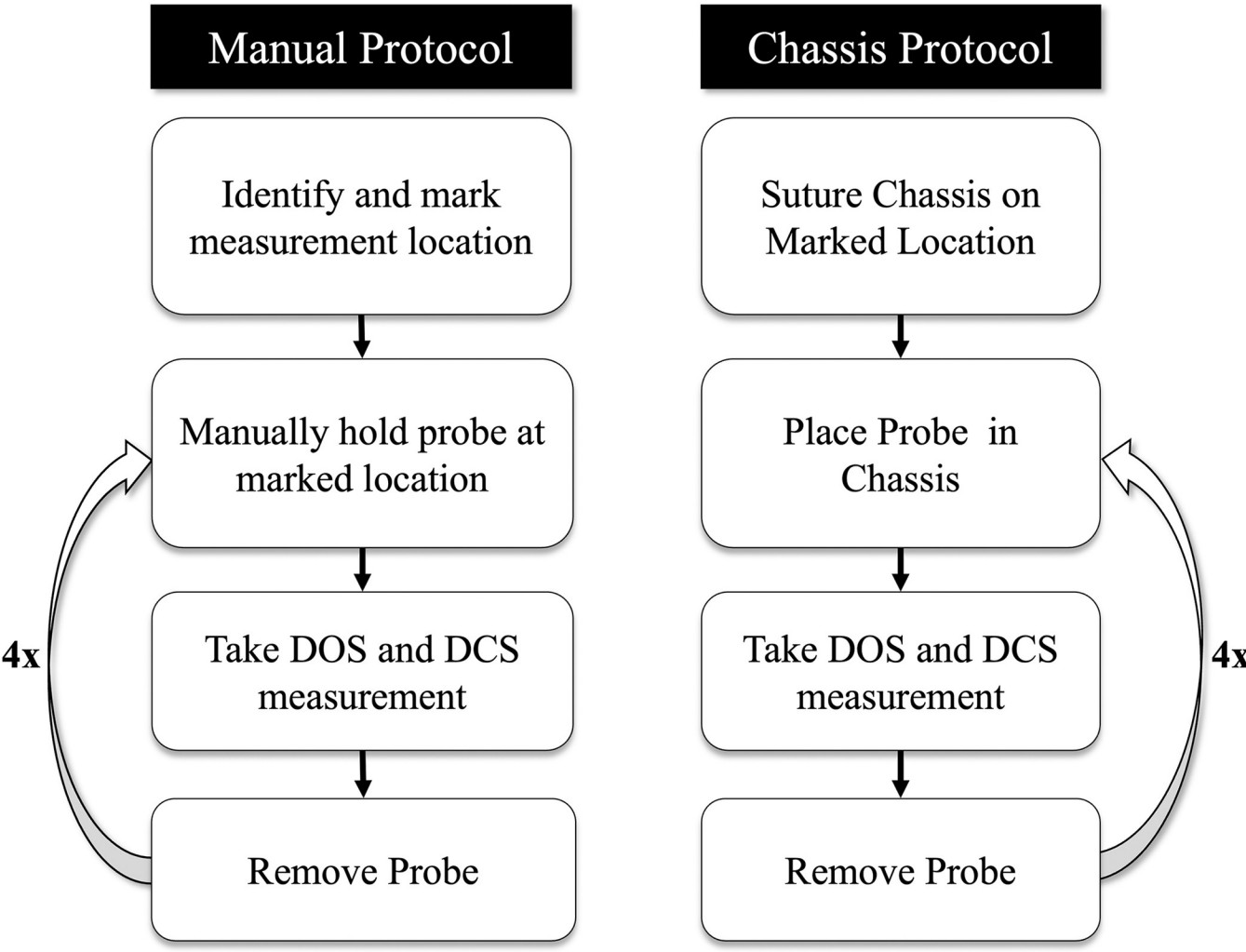

**Fig 3. Reproducibility assessment protocol.** Flow chart demonstrating the protocol used for obtaining Manual and Chassis measurements from which reproducibility was assessed and compared.

measurement within each subject using the following null ($H_0$) and alternate ($H_a$) hypotheses:

$$\text{Two−sided}: \qquad H_0 : \bar{x}_{i\,Chassis} = \bar{x}_{i\,Manual} \quad H_a : \bar{x}_{i\,Chassis} \neq \bar{x}_{i\,Manual} \tag{1}$$

Here, $\bar{x}_{i\,Chassis}$ represents the mean of measurements for subject $i$ using the chassis method, and $\bar{x}_{i\,Manual}$ represents the mean of measurements for subject $i$ using the manual method.

The coefficient of variation (CV) was used to evaluate reproducibility using the equation:

$$CV_i^j = \frac{sd_i^j}{\bar{x}_i^j}, \tag{2}$$

where $\bar{x}_i^j$ and $sd_i^j$ are the mean and standard error for the measurements in a specific setting of parameter and method $j$ for patient $i$, e.g., $j = 1$ for "manual", and 2 for "chassis". The CV evaluates the variability of data in a sample in relation to the sample mean.

To compare the reproducibility of the manual versus the chassis method, a one-sided paired $t$-test and a two-sided paired $t$-test were performed using the following equation and

hypotheses:

$$\text{One-sided}: \quad H_0: CV_{Chassis} \leq CV_{Manual} \quad H_a: CV_{Chassis} > CV_{Manual} \tag{3}$$

$$\text{Two-sided}: \quad H_0: CV_{Chassis} = CV_{Manual} \quad H_a: CV_{Chassis} \neq CV_{Manual} \tag{4}$$

To compare the CV of the BFI data to the $StO_2$ data, the above method was also undertaken using the following hypotheses:

$$\text{One-sided}: \quad H_0: CV_{BFI} \leq CV_{StO2} \quad H_a: CV_{BFI} > CV_{StO2} \tag{5}$$

$$\text{Two-sided}: \quad H_0: CV_{BFI} = CV_{StO2} \quad H_a: CV_{BFI} \neq CV_{StO2} \tag{6}$$

*P*-values were obtained using permutation for all CV comparisons, where the empirical null distribution of the test statistics was obtained by permuting the method labels (e.g., manual vs. chassis or BFI vs. $StO_2$) 1000 times. Statistical significance was assessed for all analyses using a significance level of 0.05.

## Results

### Chassis-based optical probe design and assembly

The final chassis probe design was comprised of three individually 3D-printed pieces; two pieces formed the fiber housing probe body, and one was the chassis itself. The probe body was composed of a top and bottom piece, where the bottom piece maintained fixed source-detector separations, and a separate top piece applied downward pressure to secure optical fibers in place following insertion into the bottom piece. The assembled probe body was then designed to fit securely into the circumferential chassis. Fig 4 depicts the historical probe design (Fig 4A) versus the final chassis probe design (Fig 4B).

Two different 3D printed materials were tested: Formlabs tough resin (FLTOTL05, Formlabs, Somerville, Massachusetts) and Formlabs black resin (FLGPBL04, Formlabs, Somerville, Massachusetts). The tough resin probe (Fig 4B) was more durable, however, the material color and transparency allowed light contamination that affected signal quality. Therefore, the black resin (Fig 4C) was chosen for our final design due to its opacity and durability.

Individual pieces of the final probe are shown in Fig 5A. Corresponding divots and magnets installed within the probe body and within the chassis facilitated rapid alignment during insertion. After insertion, thumb screws inserted through the chassis and terminating in the probe body ensured rigid fixation to prevent translation of the probe body within the chassis. The chassis featured strategically positioned suture points for secure attachment and ease of

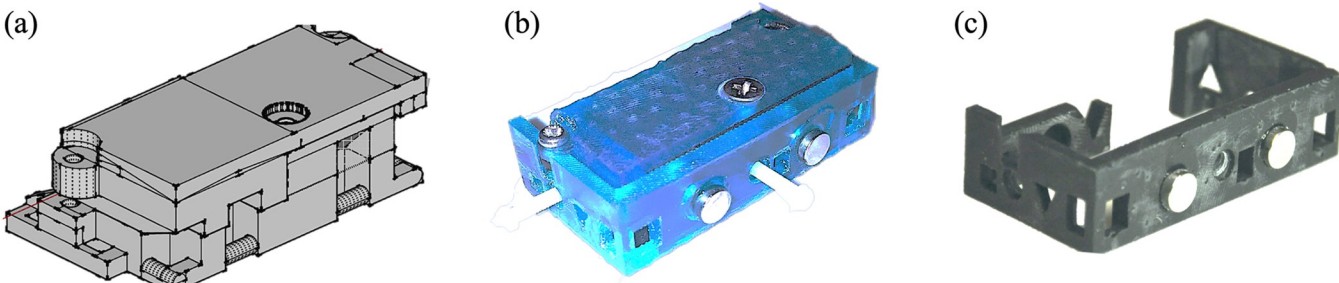

**Fig 4. Design iterations.** (a) Historical probe design with embedded suture points. (b) The chassis probe design printed in Formlabs tough resin and assembled. (c) The stand-alone chassis piece printed in Formlabs black resin.

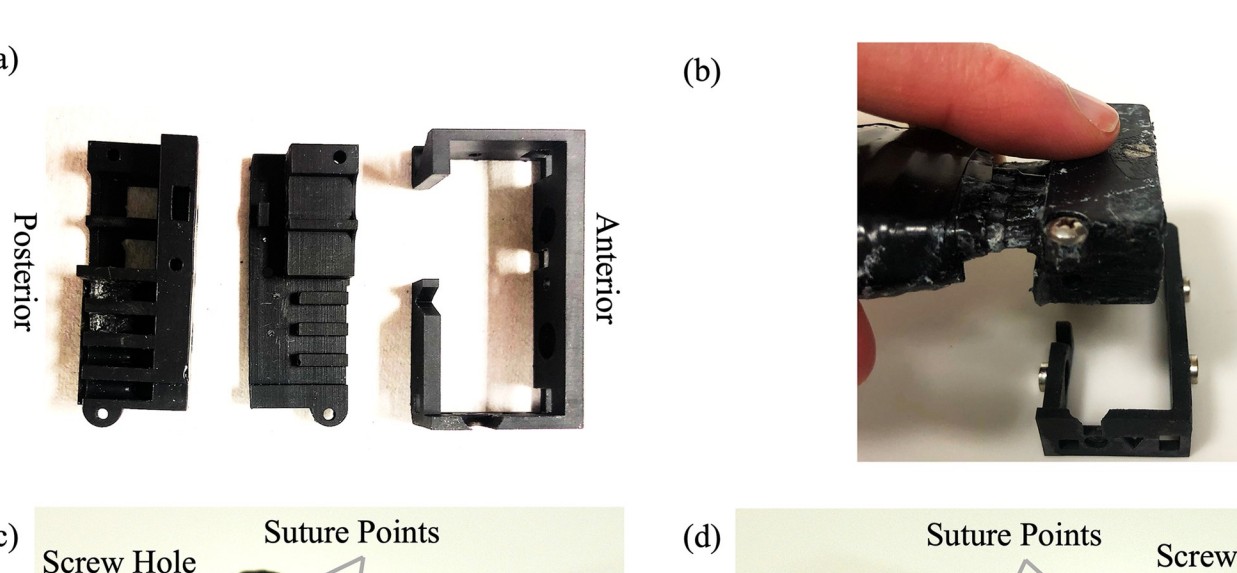

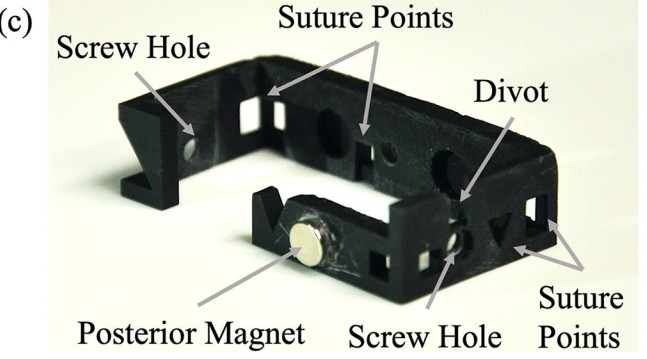

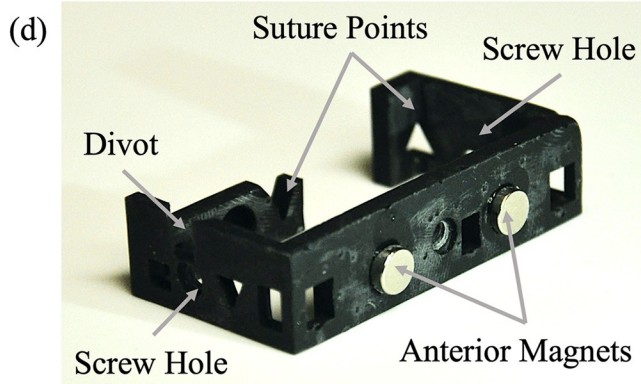

**Fig 5. Visualization of probe.** (a) From left to right, the top and bottom pieces of the probe body and chassis. (b) The assembled probe body being inserted into the chassis. (c, d) The chassis design with features indicated by gray arrows.

suturing. The chassis and all its features, including magnets, divots, and screw holes, can be seen in a posterior and anterior view in Fig 5C and 5D, respectively.

## Design validation and assessment of measurement reproducibility

After assembling the final probe design, we compared the accuracy and reproducibility of DCS measurement of BFI and FD-DOS measurement of $StO_2$, THC, $[HbO_2]$, and $[Hb]$ collected using the chassis probe versus manual probe alignment in 20 pediatric swine. The results of individual measurement repetitions within each subject using manual versus chassis probe alignment are shown for each parameter in Fig 6.

The within-subject coefficient of variation (CV) of measurement repetitions is shown for each subject in Fig 7.

The across-subject mean and standard deviation of the within-subject mean and coefficient of variation values for each parameter are summarized in Table 1.

No significant differences (p>0.05) were observed between mean values acquired using the chassis method versus the manual method for all evaluated parameters. This demonstrates comparable measurement accuracy between methods. The average coefficient of variation (CV) within subjects for the manual versus chassis for each measurement of BFI, $StO_2$, THC, $[HbO_2]$, and $[Hb]$ were also compared for each parameter using one-sided and two-sided t-tests (Fig 7, Table 1). Both t-tests did not detect a statistically significant difference in

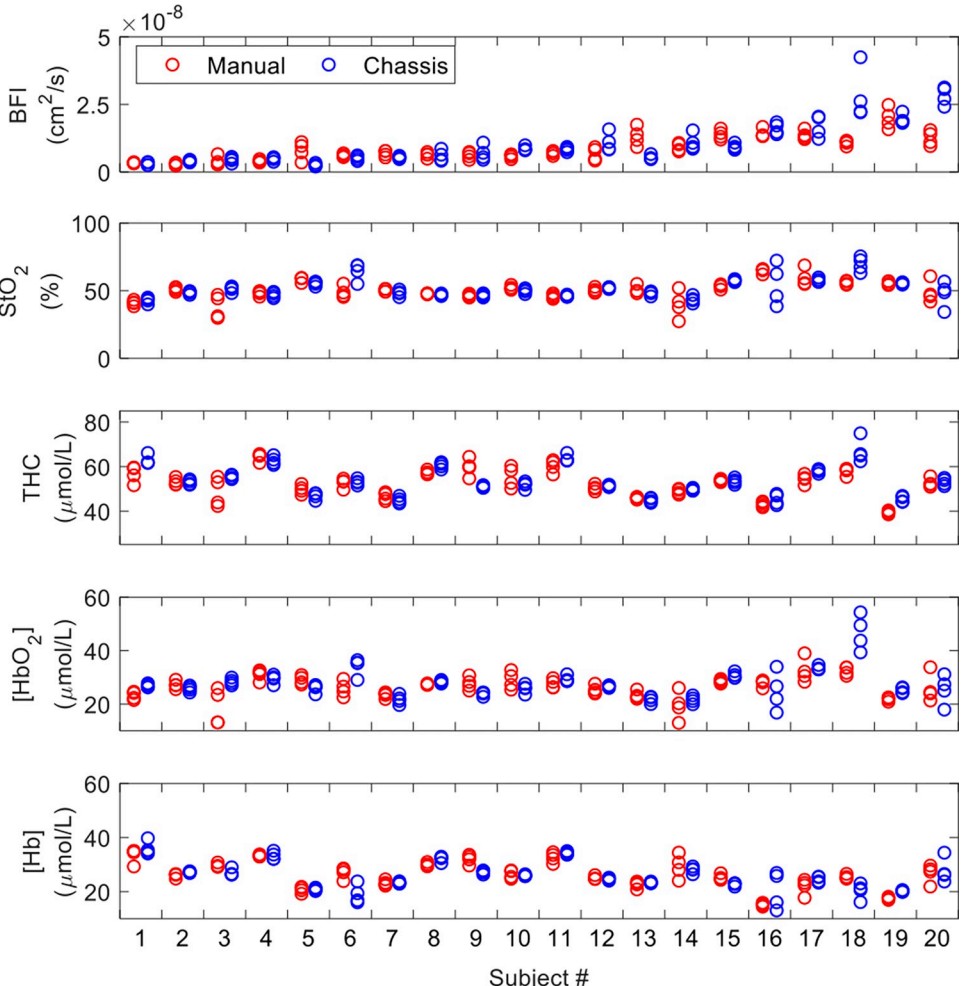

**Fig 6. Measurement data.** A scatter plot of BFI, $StO_2$, THC, $[HbO_2]$, and [Hb] measurement values (four iterations) for all subjects (plotted along the x-axis) for each method, either manual (*red*) or chassis *(blue)*. *Abbreviations*: BFI, blood flow index; $StO_2$, tissue oxygen saturation; THC, Total Hemoglobin Concentration; $[HbO_2]$, Concentration of Oxygenated Hemoglobin; [Hb], Concentration of Deoxygenated Hemoglobin.

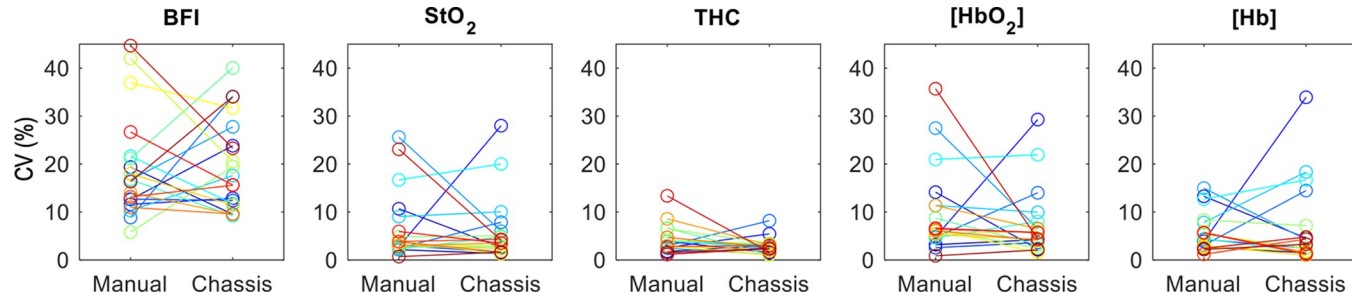

**Fig 7. Average CV data.** Plot of the average Coefficient of Variation (CV, %) for each subject (as indicated by separate color) by measurement method for BFI, $StO_2$, THC, $[HbO_2]$, and [Hb] (the connecting line connects data from the same subject with both modalities). *Abbreviations*: BFI, blood flow index; $StO_2$, tissue oxygen saturation; THC, Total Hemoglobin Concentration; $[HbO_2]$, Concentration of Oxygenated Hemoglobin; [Hb], Concentration of Deoxygenated Hemoglobin.

**Table 1. Comparison of manual and chassis probe measurements.**

| Parameter | Comparison of Absolute Values | | | | Comparison of the Coefficient of Variation | | | | |
|---|---|---|---|---|---|---|---|---|---|
| | Manual | Chassis | t-stat | p-value (two-sided) | Manual | Chassis | t-stat | p-value (one-sided) | p-value (two-sided) |
| BFI (cm$^2$/s*10$^{-8}$) | 0.9 (0.5) | 1.0 (0.7) | -1.175 | 0.255 | 19.0% (10.8) | 19.5% (9.6) | 0.166 | 0.567 | 0.861 |
| StO$_2$ (%) | 50.3 (6.6) | 51.7 (6.8) | -0.967 | 0.346 | 6.6% (7.1) | 5.8% (6.7) | -0.429 | 0.382 | 0.725 |
| THC (μmol/L) | 52.6 (6.1) | 53.7 (6.8) | -1.237 | 0.231 | 4.1% (2.9) | 2.8% (1.6) | -1.595 | 0.062 | 0.129 |
| [HbO$_2$] (μmol/L) | 26.4 (3.8) | 27.8 (5.7) | -1.348 | 0.194 | 9.7% (8.8) | 7.3% (7.0) | -0.972 | 0.168 | 0.364 |
| [Hb] (μmol/L) | 26.2 (5.1) | 25.9 (5.0) | 0.4 | 0.693 | 5.7% (3.9) | 6.5% (8.3) | 0.366 | 0.61 | 0.76 |

*Abbreviations*: BFI, blood flow index; StO$_2$, tissue oxygen saturation; THC, Total Hemoglobin Concentration; [HbO$_2$], Oxygenated Hemoglobin; [Hb], Deoxygenated Hemoglobin.

reproducibility ($p<0.05$) for any parameter. The lack of significance of the one-sided test indicates that the reproducibility of the chassis method did not exceed that of the manual method. Additionally, the lack of significance of the two-sided test indicates that reproducibility was comparable between methods. THC data exhibited a trend towards lower chassis CV values compared to manual CV values (one-sided p-value = 0.062) indicating a potential reproducibility improvement using the chassis method.

We performed a secondary analysis to compare measurement reproducibility (i.e., CV) between StO$_2$ and BFI, two commonly used physiologic diffuse optics metrics (Fig 8). We observed that the BFI data had the highest CV values, followed by StO$_2$ and THC. Both a one-sided and a two-sided t-test found that the CV values of the BFI data were significantly greater than the StO$_2$ data CV ($p<0.001$).

## Discussion

Optical neuromonitoring is an emerging clinical modality that may supply clinical care providers with real-time information on cerebral health at the bedside in critical situations like cardiopulmonary resuscitation (CPR) to inform clinical management. The presented novel chassis-based probe makes strides to address challenges of rapid, reproducible mechanical attachment to facilitate longitudinal monitoring of these critical scenarios. The probe design enables diffuse optical measurements while incorporating specific design considerations for rapid probe attachment/detachment and reproducible re-attachment, durability, and cost. We demonstrate its implementation in a preclinical setting. Application of the novel chassis-probe in a large animal model validated that the reproducibility of measurements following detachment and re-attachment were comparable to manual hand-held measurements with visual landmarks.

It is important to note that due to the use of visual landmarks, the manual hand-held measurements in our study represent optimal reproducibility with respect to probe localization. Thus, it is significant that after probe detachment, rapid chassis-based probe re-attachment without visual alignment resulted in reproducibility that did not significantly differ from the optimal manual method. Our reproducibility results agree with the hybrid time-resolved DOS/DCS device used in Andresen et al. for measurement of StO$_2$ (intra-subject CV = 5.7%; 5.8–6.6% in our study) and showed moderate improvement for measurement of BFI (intra-subject CV = 27%; 19.0–19.5% in our study) [35]. Commercially available NIRS devices have

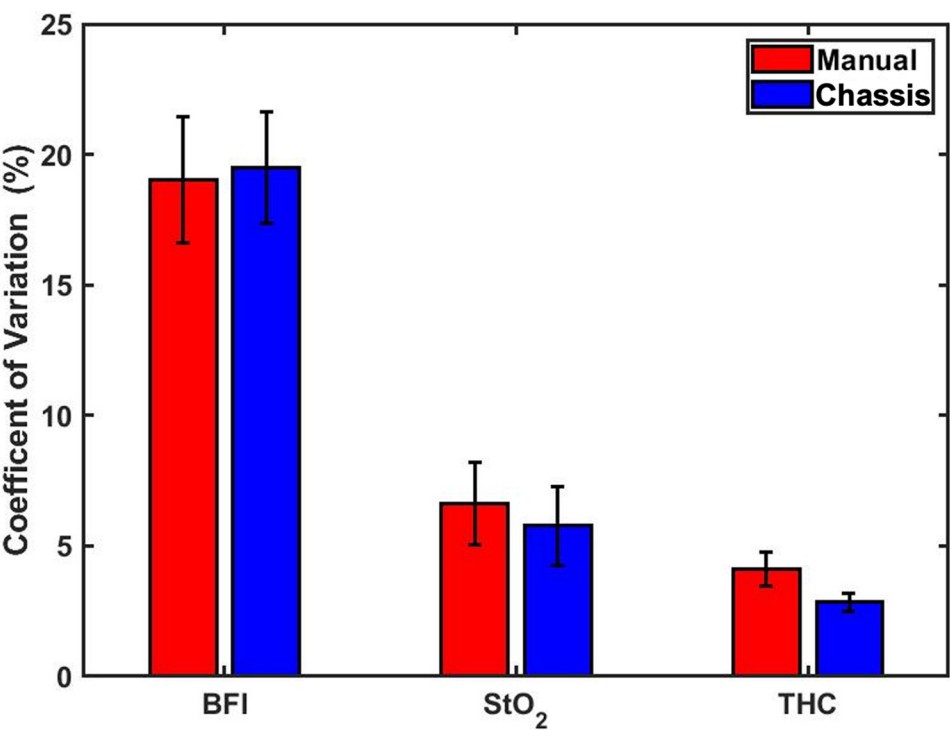

**Fig 8. CV standard error of mean.** Bar Plot of the Coefficient of Variation (%) with error bars representing the standard error of the mean (SEM) of BFI, $StO_2$, THC for each subject for all four measurements in both manual (*red*) and chassis (*blue*) methods. No statistically significant differences in CV were observed between methods. *Abbreviations*: BFI, blood flow index; $StO_2$, tissue oxygen saturation; THC, Total Hemoglobin Concentration.

also reported an $StO_2$ intra-subject reproducibility coefficient between 6–7% on the heads of healthy resting newborns [45]. The comparability of commercially available NIRS as well as other advanced research DOS/DCS devices partially reflects intrinsic measurement precision and signal-to-noise ratio (SNR), which is impacted by the detected light intensity. Our finding that the reproducibility of DCS-measured BFI is significantly lower than DOS-measured $StO_2$ agrees with Andresen *et al*. This discrepancy is an important factor to consider in experimental planning to ensure adequate statistical power is achieved for both parameters. In the future, advancements in hardware and software to improve SNR will result in further improvements in reproducibility.

Probe design for secure subject attachment is a persisting design challenge for continuous diffuse optical neuromonitoring which has been previously addressed in numerous ways. A non-exhaustive, representative survey of examples of non-invasive fiber optic probes used in DCS-only or DCS/DOS measurements of the brain in large animals or human subjects is compiled in Table 2 by modality and highlights various probe materials and attachment schema. Across all modalities, a headwrap was most commonly used to attach the optical probe to the head. This schema applies relatively uniform downward pressure across the probe to ensure skin contact and provides material coverage around the perimeter of the probe to prevent ambient light contamination. However, it is challenging to reproducibly apply; variability in anatomical coverage, optical probe coverage, and headwrap tension could impact measurement reproducibility. Critically, on its own, a headwrap does not maintain probe location after detachment.

An alternative headframe attachment schema which partially addresses this limitation has also been utilized, though in fewer studies [28, 46–49]. In Mesquita, et al. [46], the optical

**Table 2. Examples of fiber-coupled diffuse optical probe materials and attachment methods by modality.**

| Probe Material | DCS only | | DOS + DCS | |
|---|---|---|---|---|
| Rigid | Li, 2005 [47] | | Lee, 2022 [50]<br>Rehberger, 2015 [51] | |
| Flexible | | | | |
| Perforated Rubber | | | Poon, 2020 [52]<br>Roche-Labarbe, 2010 [53] | |
| Molded Rubber | Favilla, 2023 [54]<br>Dar, 2020 [55]<br>Mullen, 2019 [56]<br>Wang, 2019 [57] | | Tabassum, 2023 [58]<br>Shaw, 2023 [59]<br>Forti, 2020 [60]<br>He, 2018 [61] | Delgado., 2018 [62]<br>Buckley, 2013 [63]<br>Roche., 2009 [64] |
| Foam Sheet | Baker, 2017 [65]<br>Durduran, 2014 [23] | Mesquita, 2013b [46]<br>Buckley, 2009 [8] | Mesquita, 2013a [66]<br>Durduran, 2010 [67] | |
| Flexible 3D-Printed | Wu, 2021 [68] | | Bili, 2023 [69]*<br>Harvey., 2023 [70]*<br>Zavriyev, 2021 [71]<br>Rajaram, 2018 [48]*<br>Rajaram, 2020 [28] | Milej, 2020 [49]<br>Tamborini, 2019 [72]<br>Carp, 2017 [73]<br>Lin, 2016 [74] |
| Flexible polyimide | Renna, 2022 [31] | | | |
| **Attachment Methods** | DOS only | | DOS + DCS | |
| Hand-held | Zhao, 2023 [75] | | Lee, 2022 [50]<br>Lin, 2016 [74] Buckley, 2013 [63] | Roche., 2010 [53]<br>Roche., 2009 [64] |
| Frame | Mesquita, 2013b [46]<br>Li, 2005 [47] | | Rajaram, 2020 [28]<br>Rajaram, 2018 [48]* | Milej, 2020 [49] |
| Headwrap | Favilla, 2023 [54]<br>Feng, 2023 [76]<br>Udina, 2022 [77]<br>Wu, 2021 [68]<br>Mullen, 2019 [56]<br>Wang, 2019 [57] | Baker, 2017 [65]<br>Parthasar., 2017 [78]<br>Durduran, 2014 [23]<br>Mesquita, 2013b [46]<br>Buckley, 2009 [8] | Shaw, 2023 [59]<br>Harvey., 2023 [70]*<br>Tabassum, 2023 [58]<br>Poon, 2020 [52]<br>Forti, 2020 [60]<br>Forti, 2019 [5] | Tamborini, 2019 [72]<br>Carp, 2017 [73]<br>Mesquita, 2013a [66]<br>Durduran, 2010 [67]<br>Durduran, 2004 [79] |
| Tape | Favilla, 2023 [54]<br>Udina, 2022 [77]<br>Ozana, 2021 [80] | Dar, 2020 [55]<br>Durduran, 2014 [23] | Harvey., 2023 [70]*<br>Tabassum, 2023 [58]<br>Zavriyev, 2021 [71]<br>Rajaram, 2020 [28] | Forti, 2019 [5]<br>He, 2018 [61]<br>Delgado., 2018 [62] |
| Hydrogel | Renna, 2022 [31] | | | |

*Abbreviations*: DCS, diffuse correlation spectroscopy; DOS, diffuse optical spectroscopy

\* Probes used in Preclinical Studies

probe was secured to the head by an overlying rigid headframe that is used clinically for transcranial doppler ultrasound monitoring. The head frame featured both an adjustable circumferential and coronal band. While probe re-attachment was not necessary in this prior study, reproducible probe placement could be achieved by attaching the probe to the frame. With the use of anatomical landmarks for frame placement, both reproducible frame and probe re-attachment may be achieved. However, similar to visual re-alignment of the probe alone, the time and attention necessary for headframe placement and alignment may also impede use in time-sensitive scenarios. A flexible 3D-printed frame combined with an elastic head band, used in Rajaram, et al. [28] and Milej, et al. [49], is most similar in concept to our chassis-based probe design. The frame permits dynamic detachment and re-attachment of individual fiber bundles to facilitate patient mobility, and the additional use of tape prevented translation of the headframe across the skin surface. However, detachment via removal of individual fibers poses a challenge to the reproducibility of the FD-DOS technique, specifically, where calibration constants can be invalidated by even minor alterations in fiber orientation and position.

Our detachable circumferential chassis features the combined benefits of maintaining precise source-detector fiber positioning within the detachable probe body while also providing reproducible localization via chassis attachment. The compact form factor of the chassis may be rapidly applied and feasible to leave in-place between discontinuous monitoring sessions. However, we anticipate the need for additional design revisions to achieve clinical implementation. Most obviously, suture attachment of the chassis must be replaced with an alternative method; a promising solution is the use of double-sided, skin-safe adhesives. To facilitate adhesion, an expanded footprint may be necessary. Additionally, we anticipate the need to examine alternative probe materials. Our study utilized simultaneous DCS and multi-distance FD-DOS monitoring; existing clinical probes that have utilized these modalities are commonly made of molded rubber or flexible 3D printed materials. This adaption of our rigid preclinical probe design may be necessary for patient comfort and to maintain probe contact in young patients with high skin surface curvature. In the future, successful clinical implementation holds promise to improve longitudinal reproducibility and reduce the time, personnel, and expertise required for monitoring initiation.

## Limitations

Assessment of the measurement reproducibility of the novel chassis probe design was performed in comparison to a manually aligned, hand-held probe using visual markers, reflecting a "best case" scenario. Comparison to manual hand-held measurements where the measurement location is not demarcated may result in a significant difference in reproducibility compared to the chassis probe. There are several additional factors that were not evaluated in this study that may demonstrate additional utility of the chassis paradigm. For example, future studies should assess the stability of probe attachment (using contact and pressure sensors), time required to re-attach a probe, and tolerance to head motion using the chassis versus other methods in human patients. Clinical translation of our design from a suture-based chassis attachment to an adhesive-based chassis attachment will require the proper selection of adhesives, with a focus on durability and skin-safety. In addition, modifications of the chassis will be required to ensure that the adhesives do not impair proper coupling of the probe to the chassis and the skin surface.

## Conclusions

The novel 3D printed chassis-probe design presented in this work can perform reproducible non-invasive diffuse optical measurements of cerebral hemodynamics following detachment and re-attachment of the probe from the measurement location. Furthermore, the probe design is cost-effective and maintains integrity during suture attachment in preclinical measurement settings. The reproducibility of this chassis design is comparable to existing solutions and is a promising paradigm to address challenges of reproducible probe placement for longitudinal monitoring in clinical populations.

## Supporting information

**S1 File. Raw data.** This excel spreadsheet contains the raw DCS and DOS data from the experiments that were analyzed in this article. It is separated by subject as well as by measurement type and measurement number.
(XLSX)

## Acknowledgments

The authors would like to thank the University of Pennsylvania Libraries' Biomedical Library for their 3D printing services and the veterinary staff at the Children's Hospital of Philadelphia for their attentive care for the welfare of our animals. We would also like to thank Dr. David H. Jang from the Poison Control Center at Children's Hospital of Philadelphia and Dr. Misun Hwang from the Department of Radiology at Children's Hospital of Philadelphia for their assistance.

## Author Contributions

**Conceptualization:** Giselle C. Matlis, Emilie J. Benson, Brian R. White, Tiffany S. Ko.

**Data curation:** Giselle C. Matlis, Qihuang Zhang, Emilie J. Benson, Kristen Andersen, Jharna Jahnavi, Jake Breimann, Rodrigo M. Forti, Wesley B. Baker, Rui Xiao, Tiffany S. Ko.

**Formal analysis:** Giselle C. Matlis, Qihuang Zhang, Jharna Jahnavi, Jake Breimann, Rui Xiao, Tiffany S. Ko.

**Funding acquisition:** Daniel J. Licht, Arjun G. Yodh, Todd J. Kilbaugh, Brian R. White, Wesley B. Baker, Rui Xiao, Tiffany S. Ko.

**Investigation:** Giselle C. Matlis, Emilie J. Benson, Kristen Andersen, Jharna Jahnavi, Alec Lafontant, Jake Breimann, Thomas Hallowell, Yuxi Lin, Wesley B. Baker, Tiffany S. Ko.

**Methodology:** Giselle C. Matlis, Qihuang Zhang, Emilie J. Benson, M. Katie Weeks, Alec Lafontant, Thomas Hallowell, Yuxi Lin, Todd J. Kilbaugh, Rodrigo M. Forti, Brian R. White, Wesley B. Baker, Rui Xiao, Tiffany S. Ko.

**Project administration:** Giselle C. Matlis, Wesley B. Baker, Tiffany S. Ko.

**Resources:** Kristen Andersen, Jharna Jahnavi, Jake Breimann, Daniel J. Licht, Arjun G. Yodh, Todd J. Kilbaugh, Wesley B. Baker, Rui Xiao, Tiffany S. Ko.

**Software:** Giselle C. Matlis, Emilie J. Benson, Jharna Jahnavi, Jake Breimann, Wesley B. Baker, Tiffany S. Ko.

**Supervision:** Daniel J. Licht, Arjun G. Yodh, Todd J. Kilbaugh, Wesley B. Baker, Rui Xiao, Tiffany S. Ko.

**Validation:** Giselle C. Matlis, Emilie J. Benson, Rodrigo M. Forti, Brian R. White, Wesley B. Baker, Tiffany S. Ko.

**Visualization:** Giselle C. Matlis, Emilie J. Benson, Wesley B. Baker, Tiffany S. Ko.

**Writing – original draft:** Giselle C. Matlis, Tiffany S. Ko.

**Writing – review & editing:** Giselle C. Matlis, Qihuang Zhang, Emilie J. Benson, M. Katie Weeks, Kristen Andersen, Jharna Jahnavi, Alec Lafontant, Jake Breimann, Thomas Hallowell, Yuxi Lin, Daniel J. Licht, Arjun G. Yodh, Todd J. Kilbaugh, Rodrigo M. Forti, Brian R. White, Wesley B. Baker, Rui Xiao, Tiffany S. Ko.

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
