## [Decision Letter · Decision Letter 0]

3 Apr 2024

PONE-D-24-07227Chassis-based fiber-coupled optical probe design for reproducible quantitative diffuse optical spectroscopy measurementsPLOS ONE

Dear Dr. Ko,

Thank you for submitting your manuscript to PLOS ONE. After careful consideration, we feel that it has merit but does not fully meet PLOS ONE’s publication criteria as it currently stands. Therefore, we invite you to submit a revised version of the manuscript that addresses the points raised during the review process.

 **There are only minor requests to be addressed.**

We look forward to receiving your revised manuscript.

Kind regards,

Alberto Dalla Mora, Ph.D.

Academic Editor

PLOS ONE

2. We note that Figures 1C, 4 and 5 in your submission contain copyrighted images. All PLOS content is published under the Creative Commons Attribution License (CC BY 4.0), which means that the manuscript, images, and Supporting Information files will be freely available online, and any third party is permitted to access, download, copy, distribute, and use these materials in any way, even commercially, with proper attribution. For more information, see our copyright guidelines: http://journals.plos.org/plosone/s/licenses-and-copyright.

1. You may seek permission from the original copyright holder of Figures 1C, 4 and 5 to publish the content specifically under the CC BY 4.0 license.

Reviewers' comments:

Reviewer's Responses to Questions

**Comments to the Author**

1. Is the manuscript technically sound, and do the data support the conclusions?

Reviewer #1: Yes

Reviewer #2: Yes

2. Has the statistical analysis been performed appropriately and rigorously? 

Reviewer #1: Yes

Reviewer #2: Yes

3. Have the authors made all data underlying the findings in their manuscript fully available?

Reviewer #1: Yes

Reviewer #2: Yes

4. Is the manuscript presented in an intelligible fashion and written in standard English?

Reviewer #1: Yes

Reviewer #2: Yes

5. Review Comments to the Author

Reviewer #1: The authors present the study aimed at developing and practical testing a novel chassis-based optical probe design for DOS/DCS measurements and validating its accuracy and reproducibility against conventional, manually held measurements of cerebral hemodynamics. In this work, pediatric swine was used to validate optical probe performance. The authors demonstrate agreement in performance between manual alignment and the chassis-based DOS/DCS probe design. It should be emphasized that chassis-based probes facilitate rapid probe attachment/reattachment, which, in many instances, is crucial and very useful for practical applications. It also provides stable reproducibility conditions since the probe is always placed in the same spot, which would play an important role in patient monitoring over extensive periods. This is a well-written manuscript in which the conclusions are well-supported by the data presented.

I only have the following comments regarding manuscript quality. Unfortunately, many figures in the manuscript have a very poor resolution (specifically the ones showing chassis construction). The manuscript would benefit by incorporating clear high resolution images of the chassis-based probe since it will be for sure interesting for other people working in the field.

Reviewer #2: The authors presented a new chassis-based optical probe for DOS measurements, and they tested its performances (in terms of accuracy and reproducibility) on a paediatric swine. 20 swine were measured performing 8 repetitions with a conventional manual probe, and with the new chassis-based one. The reproducibility measurements show no differences between the two probes, allowing to perform measurements facilitating the probe attachment without affecting performance of the DOS device.

As reported by the authors in the discussions, different papers have been presented previously exploiting 3D-printed probes. Some of them are even more complex, as they hold different sensors. However, the probe presented in this paper sounds interesting for the particular use on preclinical study and might help in retrieving more accurate hemodynamic parameters. The paper is well written, and the discussions are robust.

Few minor comments:

• It is surprising that using the chassis-mode the reproducibility is similar to the manual one and it decreases only for THC. The authors highlighted that when the probe was manually positioned, a marker was facilitating the procedure, reducing the CV. However, can you comment: why you do not see this improvement only on THC? Moreover, compared to the cited paper I was expecting an enhancement in the CV.

• It is not clear to me if the probe make use of optical prisms, or mirror to deflect the light of 90°? Are the fibers perpendicular or parallel to the tissue?

• In the abstract, line 55: “Of the eight measurements, four were acquired with by placing the probe into a secured chassis, and four were visually aligned and manually held.”, please remove “with”

6. PLOS authors have the option to publish the peer review history of their article (what does this mean?). If published, this will include your full peer review and any attached files.

Reviewer #1: No

Reviewer #2: No

---

## [Author Response · Author response to Decision Letter 0]

21 May 2024

We thank the editors and reviewers for their thorough review of our original manuscript and determination that “there are only minor requests to address”. Please find enclosed our revised manuscript for your reconsideration for publication in PLOS ONE as an original research article.

We are pleased that both reviewers found merit in this work, specifically Reviewer 1 summarized our work as “a well-written manuscript in which the conclusions are well-supported by the data presented,” and Reviewer 2 wrote that, “the paper is well written, and the discussions are robust.” We have carefully considered each reviewer’s comments and performed succinct revisions to address each point. Specifically, Reviewer 1 commented on image quality which has been improved, and Reviewer 2 requested additional methodologic details which have been added to the text. The improvements have added clarity and strengthened our original manuscript.

Additionally, the editor raised concerns regarding potential copyright of three images within our manuscript (Figure 1c, Figure 4, and Figure 5). These images are all original works that, to the best of our knowledge, have not been previously published and do not require licensing. A Google reverse image search did not yield any image matches. Thus, no attributions statements were added.

We believe our responses below and revisions should satisfy the concerns of the editors and reviewers. Thank you for your time and consideration.

Reviewer 1, Comment 1: Unfortunately, many figures in the manuscript have a very poor resolution (specifically the ones showing chassis construction). The manuscript would benefit by incorporating clear high resolution images of the chassis-based probe since it will be for sure interesting for other people working in the field.

Response: Thank you for bringing this to our attention. We have revised all figures to improve image resolution and clarity.

Changes to text: Figures 1-8 have been updated and formatted according to journal guideline.

Reviewer 2, Comment 1: It is surprising that using the chassis-mode the reproducibility is similar to the manual one and it decreases only for THC. The authors highlighted that when the probe was manually positioned, a marker was facilitating the procedure, reducing the CV. However, can you comment: why you do not see this improvement only on THC? Moreover, compared to the cited paper I was expecting an enhancement in the CV.

Response: Thank you for your comment. As you astutely noted, we did have modest improvement in THC in the chassis versus manual measurements but not in other parameters (StO2 or BFI). We believe the general comparability between the two methods is a reflection of “best case” attachment and reflects well on the chassis-based measurement because the manual method was precisely aligned each time. 

Regarding the comparison of our results to prior work, our COV in BFI was ~19% and is a moderate improvement compared to prior reports of 27% in Andresen et al. We will make note of this in the discussion. The comparable COV of StO2 may reflect limitations imposed by the SNR of the StO2 measurements itself. This is influenced by the signal-to-noise ratio (SNR) of the amplitude and phase measurements of the FD-DOS measurement which is impacted by detected light intensity levels and ambient light leakage. In the future, hardware advancements which reduce measurement and quantization noise could further reduce measurement variability.

Changes to text: In “Discussion”, we added text to reflect that our BFI COV was moderately improved compared to prior work. Further we added additional discussion regarding the potential role of SNR in measurement variability and opportunities for future improvement with improved instrumentation.

Reviewer 2, Comment 2: It is not clear to me if the probe make use of optical prisms, or mirror to deflect the light of 90°? Are the fibers perpendicular or parallel to the tissue?

Response: Thank you for your astute question. Yes, we used optical prisms and 90-degree bend fibers to deflect light at a 90-degree angle.

Changes to text: In Methods, we clarified the use of prism-coupled optical fiber bundles and 90-degree bend optical fiber bundles.

Reviewer 2, Comment 3: In the abstract, line 55: “Of the eight measurements, four were acquired with by placing the probe into a secured chassis, and four were visually aligned and manually held.”, please remove “with”

Response: Thank you for catching this typo. We have corrected the text.

Changes to text: “with” removed in the abstract.

---

## [Decision Letter · Decision Letter 1]

28 May 2024

Chassis-based fiber-coupled optical probe design for reproducible quantitative diffuse optical spectroscopy measurements

PONE-D-24-07227R1

Dear Dr. Ko,

We’re pleased to inform you that your manuscript has been judged scientifically suitable for publication and will be formally accepted for publication once it meets all outstanding technical requirements.

Kind regards,

Alberto Dalla Mora, Ph.D.

Academic Editor

PLOS ONE

Additional Editor Comments (optional):

Reviewers' comments:

Reviewer's Responses to Questions

**Comments to the Author**

1. If the authors have adequately addressed your comments raised in a previous round of review and you feel that this manuscript is now acceptable for publication, you may indicate that here to bypass the “Comments to the Author” section, enter your conflict of interest statement in the “Confidential to Editor” section, and submit your "Accept" recommendation.

Reviewer #2: All comments have been addressed

2. Is the manuscript technically sound, and do the data support the conclusions?

Reviewer #2: Yes

3. Has the statistical analysis been performed appropriately and rigorously? 

Reviewer #2: Yes

4. Have the authors made all data underlying the findings in their manuscript fully available?

Reviewer #2: Yes

5. Is the manuscript presented in an intelligible fashion and written in standard English?

Reviewer #2: Yes

6. Review Comments to the Author

Reviewer #2: (No Response)

7. PLOS authors have the option to publish the peer review history of their article (what does this mean?). If published, this will include your full peer review and any attached files.

Reviewer #2: No

---

## [Editor Report · Acceptance letter]

21 Jun 2024

PONE-D-24-07227R1 

PLOS ONE

Dear Dr. Ko, 

I'm pleased to inform you that your manuscript has been deemed suitable for publication in PLOS ONE. Congratulations! Your manuscript is now being handed over to our production team.

Kind regards, 

on behalf of

Prof. Alberto Dalla Mora 

Academic Editor

PLOS ONE